# Sleeping sound with autism spectrum disorder (ASD): study protocol for an efficacy randomised controlled trial of a tailored brief behavioural sleep intervention for ASD

Nicole Papadopoulos,[1] Emma Sciberras,[1,2,3] Harriet Hiscock,[2,4] Katrina Williams,[2,3,4] Jane McGillivray,[1] Cathrine Mihalopoulos,[1] Lidia Engel,[1] Matthew Fuller-Tyszkiewicz,[1] Susannah T Bellows,[1] Deborah Marks,[1,4] Patricia Howlin,[5,6] Nicole Rinehart[1]

For numbered affiliations see end of article.

**Correspondence to**
Dr Nicole Papadopoulos;
nicole.papadopoulos@deakin.edu.au

## ABSTRACT

**Introduction** Sleep problems are a characteristic feature of children with autism spectrum disorder (ASD) with 40% to 80% of children experiencing sleep difficulties. Sleep problems have been found to have a pervasive impact on a child's socio-emotional functioning, as well as on parents' psychological functioning. The *Sleeping Sound ASD* project aims to evaluate the efficacy of a brief behavioural sleep intervention in reducing ASD children's sleep problems in a fully powered randomised controlled trial (RCT). Intervention impact on child and family functioning is also assessed.

**Methods and analysis** The RCT aims to recruit 234 children with a diagnosis of ASD, aged 5–13 years, who experience moderate to severe sleep problems. Participants are recruited from paediatrician clinics in Victoria, Australia, and via social media. Families interested in the study are screened for eligibility via phone, and then asked to complete a baseline survey online, assessing child sleep problems, and child and family functioning. Participants are then randomised to the intervention group or treatment as usual comparator group. Families in the intervention group attend two face-to-face sessions and a follow-up phone call with a trained clinician, where families are provided with individually tailored behavioural sleep strategies to help manage the child's sleep problems. Teacher reports of sleep, behavioural and social functioning are collected, and cognitive ability assessed to provide measures blind to treatment group. The primary outcome is children's sleep problems as measured by the Children's Sleep Habits Questionnaire at 3 months post-randomisation. Secondary outcomes include parent and child quality of life; child social, emotional, behavioural and cognitive functioning; and parenting stress and parent mental health. Cost-effectiveness of the intervention is also evaluated.

**Ethics and dissemination** Findings from this study will be published in peer-reviewed journals and disseminated at national and international conferences, local networks and online.

## Strengths and limitations of this study

► This sleep intervention consists of a brief behavioural intervention thus increasing the chances of the intervention being translated into clinical practice.
► This study is the largest randomised controlled trial to date on the efficacy of the Sleeping Sound program for children with autism spectrum disorder (ASD).
► Limitations of the study include: lack of an objective sleep measure, inability to control families' use of additional sleep treatments post-trial enrolment and lack of gold standard ASD diagnostic tools (eg, Autism Diagnostic Observation Schedule—Second Edition, Autism Diagnostic Interview—Revised) to confirm ASD diagnosis.

**Trial registration number** ISRCTN14077107 (ISRCTN registry dated on 3 March 2017).

## BACKGROUND

### Autism spectrum disorder

Autism spectrum disorder (ASD) is a neurodevelopmental disorder characterised by severe deficits in social, communication skills, and restricted and repetitive patterns of behaviour.[1] In 2015, it was estimated that ASD affected approximately 0.7% of the Australian population, which equates to approximately 164 000 Australians.[2] A diagnosis of ASD is associated with major child, familial and societal costs, with the economic burden in Australia estimated to be between $8.1 and $11.2 billion annually.[3] Psychiatric comorbidity is common in individuals with ASD, with approximately 70% meeting the criteria for another mental disorder,[1] including anxiety (84%)[4 5] and attention-deficit/hyperactivity

disorder (ADHD; 28% to 70%).[6 7] Sleep problems are also common in children with ASD and cause an additional burden for children and their families. However, large rigorous randomised controlled trials (RCTs) that report on the efficacy of treating sleep problems in this population are lacking.[8]

## Sleep problems in children with ASD: prevalence, aetiology and burden

It is estimated that 40%–80% of children with ASD experience sleep problems,[9–13] with sleep-onset delay, shorter sleep duration and frequent prolonged night waking being particularly common.[9 13–18] Other common sleep problems include bedtime resistance, sleep anxiety, daytime sleepiness, early waking, co-sleeping, low sleep efficiency and parasomnias.[9 13 16–20] Children with ASD will also often experience more than one sleep problem concurrently.[16] Furthermore, sleep problems in children with ASD have been shown to be more persistent over time compared with sleep problems in typically developing children.[10–12 21 22]

The aetiology of sleep problems in children with ASD is unclear and likely multifactorial,[12 20] although various biopsychosocial factors have been proposed.[13 23 24] Biological factors that may impact sleep in ASD include abnormalities in melatonin secretion, brain wave organisational and maturational differences, circadian-relevant genes, and arousal and sensory dysregulation.[13 20 25] Social-emotional and behavioural characteristics connected with core ASD features may also contribute to sleep difficulties.[13 25] Children with ASD often have difficulty recognising contextual and environmental cues that indicate that bedtime is approaching and may have difficulty settling if bedtime routines are not able to be fulfilled.[13 25] Co-occurring psychological factors, such as anxiety, depression and ADHD, as well as family factors including parental stress, parental mental health and home environment may also influence sleep problems.[11 13 20 26]

Several studies have linked sleep problems with behavioural difficulties and psychopathology in ASD. In a large study conducted by Goldman et al[27] (n=1784), children with ASD who were poor sleepers were found to have more behavioural problems, including problems with attention and social interactions compared with good sleepers (77.7% vs 56.1%, and 75.4% vs 53.9%, respectively). Sleep problems in children with ASD have also been associated with thinking difficulties,[28] with increased communication and social deficits and higher rates of stereotypic behaviours,[29 30] and with anxiety and mood disturbances.[31] In addition, overall severity of sleep problems has been found to be associated with poorer child physical and psychosocial health-related quality of life,[32] and cognitive and motor functioning.[33]

Sleep problems have also been identified as having a pervasive impact on the family, including an elevated risk for parenting stress. For example, several studies have identified moderate to small associations between parenting stress and sleep problems in children with

ASD.[23 34–37] Two studies have linked sleep problems in children with ASD with poor parent mental health.[36 38] One study found a moderate association between child sleep problems and maternal mental health[36]; a second reported a small to medium association between child sleep problems and parent depression.[38]

## Treating sleep problems in children with ASD

Both behavioural and pharmacological interventions are commonly prescribed to treat sleep problems in ASD; however, we have much to learn about the efficacy of these treatment approaches.[8] For some sleep disorders, behavioural sleep interventions are recommended as the first line of treatment. Gringras et al[39] conducted one of the largest melatonin sleep trials involving 146 children with neurodevelopmental disabilities (including ASD). Prior to randomisation for melatonin treatment, Gringras et al[39] provided families with a booklet outlining good sleep habits and behavioural sleep strategies. Following administration of sleep strategies for a 1-month period, 35% of the sample no longer met inclusion criteria for the trial. While this study did not solely focus on children with ASD, the results indicate the potential of a brief behavioural sleep intervention for children with ASD.

Despite the promising evidence to support the use of behavioural sleep interventions for children with ASD, few studies have examined their efficacy.[8 40] A recent systematic review of behavioural sleep interventions to treat insomnia in children with neurodevelopmental disorders identified 19 ASD studies; only 3 studies employed an RCT design.[41] Findings of the three studies are mixed. Adkins et al[42] (n=36) found no effect of a standardised information pamphlet on insomnia compared with a control (no intervention) group. The findings suggest that psychoeducation alone is insufficient to produce behavioural change and that specific tailored examples may be more useful for parents when addressing child sleep problems.

Johnson et al[43] undertook a pilot RCT (n=40) comparing a five-session individualised parent behavioural sleep training program with a control group (parent psychoeducation autism program without a behavioural sleep focus). The behavioural sleep training program involved parents receiving information about how to treat behavioural sleep problems often observed in children with ASD, as well as ways of addressing future sleep issues that are typical in ASD. Families allocated to the behavioural sleep training program reported significant improvements in sleep problems (effects small to medium) as measured by the composite sleep index of the modified version of the Simonds and Parraga Sleep Questionnaire.[44 45] However, no benefits were identified using more objective measures (eg, actigraphy), and the authors did not report any improvement in child and parent outcomes despite parent satisfaction of the manualised program being high.

Malow et al[46] assessed the efficacy of a parent sleep education program on sleep-onset delay in children

with ASD (n=80), in individual versus group (two to four people) settings. Both delivery methods were effective in reducing sleep-onset delay, with large effect. Improvement was also seen in bedtime resistance and sleep duration, with medium to large effect, and sleep anxiety and night waking, with small to medium effect. Improvements in child and family functioning were also reported, including increased child quality of life and parental sense of competence, each with small to medium effect. However, findings are limited by the lack of a treatment as usual (TAU) control group.

More recently, Hiscock *et al*[47] showed that a brief two-session parent/child intervention, *Sleeping Sound,* was highly efficacious in reducing behavioural sleep problems in children with ADHD when compared with a TAU control group. The *Sleeping Sound* intervention was tailored to the family and included behavioural strategies targeted at the child's specific sleep problems. Based on this, we examined the preliminary efficacy of the *Sleeping Sound* program in 61 children with comorbid ASD-ADHD.[48] In comparison to children with ASD-ADHD receiving TAU (n=33), children with ASD-ADHD (n=28) who received the *Sleeping Sound* intervention showed large improvements in child sleep problems, as measured by parent report of reduced total scores on the Children's Sleep Habits Questionnaire (CSHQ)[49] at 3 months post-randomisation (medium to large effect) and at 6 months post-randomisation (medium effect). The intervention group also showed greater improvements in psychosocial functioning at 3-month and 6-month follow-up, with small to medium, and medium effect, respectively. Children in the intervention group were further described as showing improved emotional functioning at 3 months by parent report, with medium to large effect, and teacher report, with medium effect.

It appears that a tailored behavioural sleep intervention may improve sleep in children with ASD; however, few rigorous RCTs have been conducted.[8 40] Further, it is unclear whether behavioural sleep interventions for children with ASD lead to improvements in other domains of functioning, such as behavioural and emotional problems, social communication, parenting stress and parent mental health. Given that our pilot findings demonstrated that a tailored brief behavioural sleep intervention for children with ASD-ADHD can improve not only sleep but also emotional, behavioural and social functioning, we have selected this intervention to test in a fully powered RCT.

### Aims and hypotheses
This study aims to examine the efficacy of a brief behavioural sleep intervention (*Sleeping Sound*) in treating sleep problems in primary school children aged 5–13 years with ASD compared with a TAU group. We have chosen a TAU control group for this trial to allow us to investigate the efficacy of the Sleeping Sound intervention for children with autism who may already be receiving community or clinical interventions.

We aim to assess whether the brief sleep intervention delivered by study clinicians is related to the following outcomes:

### Primary outcome
1. Reductions in overall child sleep problem severity at 3 months post-randomisation, including reductions in total sleep disturbances, problems initiating sleep and problems maintaining sleep.

### Secondary outcomes
2. Improvements in child and primary parent functioning at 3 and 6 months post-randomisation.

We also aim to assess whether the brief sleep intervention is cost-effective when compared with TAU.

We hypothesise that compared with a TAU comparison group, children with ASD who receive the *Sleeping Sound* intervention will show greater improvements at 3 months post-randomisation (postintervention; primary outcome time point) and at 6 months post-randomisation in the following areas:
1. Reduction in overall child sleep problems.
   a. Decreased daytime sleepiness by teacher report, and improvements in sleep hygiene and sleep disturbances by parent report;
   b. Decreased emotional and behavioural problems reported by parents and teachers;
   c. Decreased social-communicative symptoms reported by parents;
   d. Increased cognitive performance, academic achievement and school attendance;
   e. Increased child quality of life reported by parents;
   f. Decreased parent stress and mental health symptoms;
   g. Increased parent quality of life and work attendance.

We also hypothesise that the behavioural sleep intervention will be a cost-effective treatment when compared with TAU from a societal and healthcare perspective. Thus, we will compare the costs and benefits (expressed in quality-adjusted life-years (QALYs)) between the two arms of the trial over the period of the study (further details are provided below).

## METHODS AND ANALYSIS
### Overall study design
The *Sleeping Sound with ASD* study is an RCT where participants are allocated either to the sleep intervention group or a TAU group. The project will run from July 2016 to the end of 2019 when all stages of the project, from recruitment to follow-up data collection at 6 months post-randomisation, will be executed. Participant recruitment concluded in March 2019; the collection of 3- and 6-month data is ongoing. This study protocol conforms with the Standard Protocol Items: Recommendations for Interventional Trials (SPIRIT) reporting requirements,[50] and a SPIRIT timeline has been provided (figure 1).

| | STUDY PERIOD | | | | |
|---|---|---|---|---|---|
| | Enrolment | | | Post-randomization | Close out |
| **TIMEPOINT** | *Baseline* | Randomization | *3m* | *6m* | |
| **ENROLMENT:** | | | | | |
| Eligibility screening | X | | | | |
| Informed consent | X | | | | |
| Baseline parent survey[a] | X | | | | |
| Randomization | | X | | | |
| **INTERVENTIONS:** | | | | | |
| Sleeping sound intervention | | | ●———● | | |
| Treatment as usual | | | | | |
| **ASSESSMENTS:** | | | | | |
| Parent survey[a] | X | | X | X | |
| Teacher survey[a] | X | | X | X | |
| Child assessment[a] | | | | X | |
| Medicare/PBS linkage | | | | | X |
| NAPLAN linkage | | | | | X |

**Figure 1** Study timeline and figure format obtained from the SPIRIT 2013 statement. NAPLAN is a national assessment of literacy and numeracy Australian children complete in Grades 3, 5, 7 and 9. [a]Refer to Table 3 for the specific measures included. m, months; PBS, Pharmaceutical Benefits Scheme; SPIRIT, Standard Protocol Items: Recommendations for Interventional Trials.

## Participants

Participants include Victorian families of children with ASD, aged 5 to 13 years, and attending primary school at the time of recruitment. All children have a parent-reported moderate to severe sleep problem of chronic insomnia and/or delayed sleep-wake phase as defined by the International Classification for Sleep Disorders—Third Edition.[51] The intervention is designed to be feasibly delivered within the context of the Australian healthcare system; therefore, eligibility for an ASD diagnosis is based on the nationally agreed DSM-IV (Diagnostic and Statistical Manual of Mental Disorders, Fourth Edition; autistic disorder, Asperger's disorder)[52] or DSM-5 (Diagnostic and Statistical Manual of Mental Disorders, Fifth Edition; ASD) assessment criteria.[1] All children have a diagnosis of ASD which has been confirmed by consensus of diagnosis by a multidisciplinary team specialised in ASD assessment (psychologist, paediatrician/psychiatrist, speech pathologist and/or occupational therapist). Written evidence of a conclusive multidisciplinary diagnosis of ASD is sighted or confirmed by the treating paediatrician for all children enrolled in the study. Children also met clinical ASD symptomology based on a clinical cut-off score of ≥11 on the Social Communication Questionnaire Lifetime form.[53]

## Patient involvement

The intervention used in this study is based on the ADHD *Sleeping Sound* trial, during which participant feedback on the acceptability and feasibility of the study was collected.[47] There were no reported concerns from participants.[47] In this trial, we will be collecting feedback data about the program from parents.

### Recruitment: stage 1

*Recruitment through Victorian paediatric clinics.* Victorian paediatricians identified potential participants meeting inclusion criteria, whom they have seen in the last 12 months, and sent a letter inviting them to participate in the study. Study advertisements were also displayed in the waiting areas of clinical services and disseminated widely through research, clinical and community-based networks available to investigators.

### Recruitment: stage 2

The research team telephoned all primary parents who registered interest in the study to assess inclusion/exclusion criteria (approximately 25 min phone call). Eligible families were provided with an information sheet, consent form (see online supplementary appendix) and baseline

| Table 1 | Study inclusion and exclusion criteria |
|---|---|
| Inclusion criteria | ▶ Clinically confirmed diagnosis of DSM-IV autistic disorder, [52] Asperger's disorder or DSM-5 ASD.[1] Diagnosis has been confirmed by consensus of a multidisciplinary team, as indicated by paediatrician or a confirmed multidisciplinary diagnosis as cited in a clinical report.<br>▶ Aged 5–12 years or 13 years and attending primary school at the time of recruitment.<br>▶ Child meets clinical ASD symptomology based on a clinical cut-off score of ≥11 on the Social Communication. Questionnaire Lifetime form (40-item questionnaire measuring ASD symptoms).[53]<br>▶ Child's sleep problem(s) are moderate or severe by primary parent report and have been a problem for them over the past 4 weeks.[66]<br>▶ Meet diagnostic criteria for at least one of the following sleep problems as defined by the International Classification of Sleep disorders diagnostic criteria:[51] chronic insomnia (including problems initiating and maintaining sleep, early morning waking, sleep-onset association, bedtime resistance and anxiety-related insomnia) or delayed sleep–wake phase. |
| Exclusion criteria | ▶ Diagnosed with an intellectual disability by parent report.<br>▶ Have any comorbid medical (eg, nocturnal seizures, blindness) conditions that disturb regular sleep patterns or genetic (eg, Fragile X disorder) conditions associated with intellectual impairment based on parent report.<br>▶ Have suspected OSA. OSA is assessed using three OSA items from the CSHQ.[49] This scale can help identify children who may suffer from OSA. If a parent endorses items indicative of OSA, the child is contacted by the study paediatrician for further assessment. If OSA is suspected, the child is excluded from the research study and referred to appropriate clinical services. After approximately 6 months, the family is followed up to see whether they are eligible to participate in the study (ie, if OSA symptoms have resolved and they have ongoing behaviour sleep problems).<br>▶ Parents with insufficient English language proficiency to provide informed consent complete study materials and/or participate in the intervention treatment program. |

ASD, autism spectrum disorder; CSHQ, Children's Sleep Habits Questionnaire; DSM-5, Diagnostic and Statistical Manual of Mental Disorders, Fifth Edition; DSM-IV, Diagnostic and Statistical Manual of Mental Disorders, Fourth Edition; OSA, obstructive sleep apnoea.

survey by email or post. The child's primary parent was asked to complete the survey and consent form. Online surveys were completed via a secure research database REDCap.[54] Once parents provided written consent, they were asked to provide diagnostic and cognitive assessment reports where available. If optional consent was provided to contact the child's school teacher, a link to an online survey was sent to the child's nominated teacher.

Families are sent a separate hardcopy consent form for researchers to access the child's Medicare Benefits Scheme (MBS) and Pharmaceutical Benefits Scheme (PBS) data for cost-effectiveness analysis. Medicare collects information on the child's doctors' and allied health professionals' visits and the associated costs (including out-of-pocket costs). PBS collects information on prescription medications which have been recorded at pharmacies.

Participants are required to meet all inclusion criteria (see table 1). Children currently prescribed melatonin as well as any other medications (eg, Selective Serotonin Reuptake Inhibitors) are included in the trial if they continue to meet eligibility criteria. Information regarding medication is collected as part of the parent survey at the time of enrolment and at 3-month and 6-month follow-up. We have included a table documenting important changes to our inclusion/exclusion criteria after trial commencement (see table 2).

## Randomisation

The flow of participants from recruitment to follow-up assessments is displayed in figure 2. Participants are randomised to either the intervention or TAU groups by an independent research assistant once consent forms and baseline surveys have been completed. Treatment allocation is determined sequentially, according to a computer-generated block randomisation sequence with 1:1 ratio between groups, with blocks of randomly varying size (4, 6 and 8) developed by an independent statistician who is not involved in the project. The independent statistician will then upload this randomisation list into REDCap,[54] a secure research database, to ensure allocation concealment. Given the relatively fewer diagnoses of ASD among girls,[55] randomisation is stratified by gender to ensure equal representation of girls, boys and other genders (ie, transgender, intersex) in both study arms. Siblings are eligible to participate in the study provided they each meet all inclusion criteria. Families with siblings enrolled in the study are randomised in sequence based on the return of the primary parent baseline surveys, with all subsequent siblings allocated to the same treatment group. All families are sent a letter informing them of their group allocation and are not able to enrol further siblings once notified of group allocation. A study clinician then telephones intervention families to book the intervention sessions.

The TAU group will receive standard services and support currently available in the community and complete the same study assessment follow-up. If the parent consented to teacher participation, researchers approach schools for consent to contact the child's teacher and then mail the teacher a baseline survey to complete.

**Table 2** Amendments to study protocol

| | |
|---|---|
| 'Opt-out' study invitation amended to 'opt in' invitation | An 'opt-out' approach was initially used to recruit through paediatricians. Parents were sent a letter from the paediatrician advising them that the research team would call them to explain the study, confirm eligibility and invite them into the study. Parents were asked to contact the study team or their paediatrician if they *did not* wish to learn more about the study or be contacted by the research team. The paediatrician then provided the research team with the contact details of families if they did not opt out within the 2-week period specified in the letter. This recruitment method was amended to an 'opt in' study invitation letter from paediatricians. |
| Age range amended | Age range extended to include 13 year olds who attend primary school at the time of recruitment. |
| Social Communication Questionnaire Lifetime form cut-off adjusted | Clinical cut-off score lowered from ≥15 to ≥11. This measure is used to assess ASD symptomology. There is evidence in the literature to support a lower cut-off score to capture children with higher-functioning ASD and Asperger's disorder who may be missed by the use of a more stringent cut-off score.[67] |
| Vineland inclusion criteria removed | The Vineland Adaptive Behaviour Scales, Third Edition (Domain Interview Survey) was originally conducted with families over the phone as a screening tool for intellectual impairment. This was proving to be too cumbersome for families and a barrier to enrolment. It was therefore removed as a requirement for eligibility. |
| Comorbid medical conditions exclusion criteria amended | Exclusion criteria for comorbid conditions refined to medical conditions affecting sleep or associated with intellectual disability. Original criteria comorbid medical (eg, epilepsy), neuropsychiatric (eg, Tourette's) or syndromic genetic (eg, Fragile X) conditions. |

ASD, autism spectrum disorder.

### Intervention group

*Delivery of the sleep intervention.* The behavioural sleep intervention is delivered by study-employed clinicians (paediatrician or psychologist experienced in working with children with ASD). Clinicians were trained over two × 3-hour interactive sessions (by HH and ES), have a detailed program manual and meet fortnightly to ensure fidelity to the program. The intervention consists of two 50 min face-to-face consultations and one follow-up phone call at 2-week intervals. The content of the original *Sleeping Sound* ADHD intervention[47] has been adapted to children with ASD to include specific visual aids and social stories to assist in the delivery of the sleep intervention for this population.[56] Intervention sessions are held in rooms at one of our study sites (Deakin University or The Royal Children's Hospital) or the treating paediatrician's consulting rooms, based on family preference.

The behavioural sleep interventions are individually tailored for the child and the family. The strategies that suit the family situation and learning style of the child with ASD are given emphasis. Delayed sleep phase, early morning waking and prolonged night waking (eg, waking overnight where the child is unable to fall back to sleep for an hour or more) are treated with bedtime fading (ie, the child's bedtime is temporarily set later and gradually brought forward), as well as early morning light exposure. Need for parental presence at sleep time is managed for sleep-onset association using a 'camping out' method. Camping out involves parents gradually withdrawing themselves from the child's bedroom over a period of time. Parent management strategies are used for bedtime resistance and can include use of a 'bedtime pass', whereby the child can only leave the bedroom once before sleep and may be coupled with a reward system. Primary insomnia (insomnia with no other established cause) is managed using relaxation training and restricting time in bed (ie, bedtime fading or getting out of bed and doing a relaxing activity if the child cannot fall asleep). In addition to relaxation training, examples of some strategies used to manage anxiety-related insomnia include discussing fears during the day rather than just before bedtime, encouraging the child to stay in bed and rewarding brave behaviour. Visual strategies, such as social scripts and visual timetables, are used to reinforce learning.

*Session One* focuses on an assessment of the child's sleep problems, psychoeducation about normal sleep and sleep cycles, good sleep habits, and a tailored sleep management plan specific to the sleep diagnosis of the child. As the Sleeping Sound intervention is individually tailored to children and families, participants are encouraged to provide input in relation to goal setting and the interventions they would like to try to assist in managing the sleep problems. Families receive written handouts, summarising session content and complete a written management plan with the clinician. A standardised consultation record is kept for all children including the presenting sleep problem(s), possible contributors to the problem (eg, TV in bedroom), medication use, comorbidities and bedtime routines. Clinicians record the duration of each consultation, sleep problem diagnoses, family sleep management goals, handouts given to parents and management options selected. Sleep problem diagnoses made may include one or more difficulties getting to sleep (sleep anxiety, bedtime resistance, sleep association onset disorder, delayed sleep phase or insomnia) and/or night waking problems

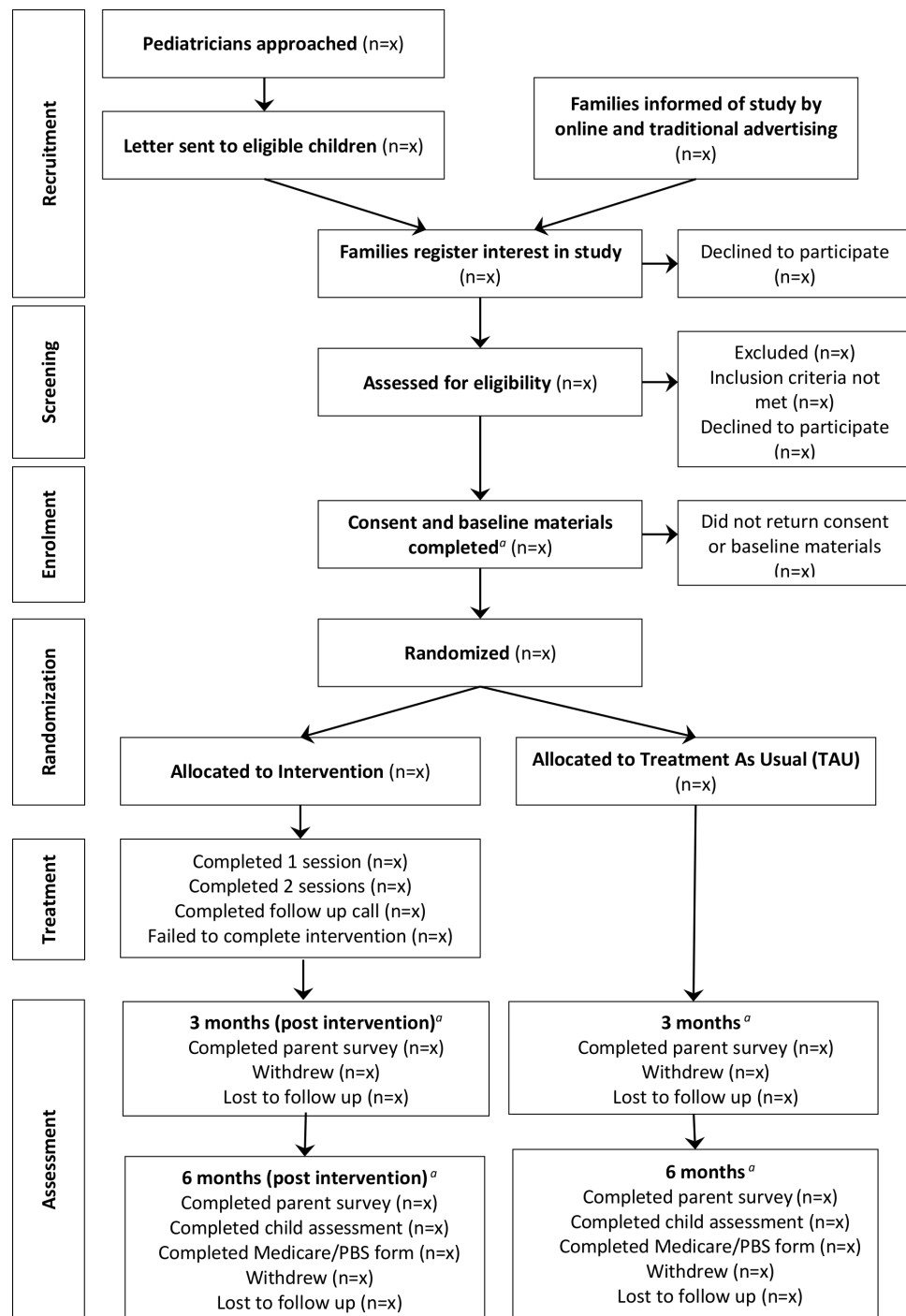

**Figure 2** Flow of participants, figure format obtained from the CONSORT (Consolidated Standards of Reporting Trials) 2010 statement.[a]Complete teacher survey if applicable.

(prolonged night waking or early morning waking). Parasomnias (nightmares, night terrors, sleep walking) are recorded, and psychoeducation and general management advice about parasomnias is provided, although these are not a focus of behavioural intervention.

*Session Two* is held 2 weeks later to review the sleep management plan (which includes a sleep diary), reinforce strategies, troubleshoot and monitor progress.

A *follow-up phone call* is conducted 2 weeks after the second visit to answer any further parent questions, reinforce strategies, troubleshoot and monitor progress.

Participants are able to withdraw from the intervention at any time. Participants are asked for feedback of the intervention at all intervention sessions, including follow-up, and this is documented by clinicians to manage any unintended adverse effects of the trial.

## Measures

Study measures are summarised in table 3 below. Additional information collected at baseline includes family and child demographics such as socioeconomic status (SES) and medical data (ie, diagnosed comorbidities).

**Table 3** Summary of measures included in the study

| Outcomes | Measure | Source | B | 3 | 6 |
|---|---|---|---|---|---|
| **CHILD OUTCOMES** | | | | | |
| Sleep problems | **Children's Sleep Habits Questionnaire (CSHQ)**. 33-item validated measure of sleep that can distinguish clinical from community samples.[49] Provides a measure of total sleep problems and eight subscale scores reflecting major behavioural sleep disorders (bedtime resistance, sleep-onset delay, sleep duration, sleep anxiety, night waking, parasomnias, sleep-disordered breathing, daytime sleepiness). | Parent | ● | ● | ● |
| Sleep hygiene | **Sleep Hygiene Scale**. 7-item study developed measure adapted from the Bedroom Routines Scale. | Parent | ● | ● | ● |
| Daytime sleepiness | **Teacher Daytime Sleepiness Questionnaire**. 10-item validated scale of daytime sleepiness at school.[68] | Teacher | ● | ● | ● |
| Social communication functioning | **Social Communication Questionnaire (SCQ) Current** form. 40-item measure of ASD symptoms in past 3 months to measure change in ASD social communication symptoms over time.[53] | Parent | ● | ● | ● |
| Emotional and behavioural disturbance | **Developmental Behavioural Checklist (DBC)**. 96-item measure of emotional and behavioural disturbance in children.[69] Provides a rating of overall behavioural disturbance and five subscales: disruptive/antisocial behaviour, self-absorbed, communication disturbance, anxiety and ASD social relating. | Parent | ● | | ● |
| Behavioural and social functioning | **Strengths and Difficulties Questionnaire (SDQ)**. 25 items assessing the following subscales: hyperactivity/inattention, conduct problems, emotional symptoms, peer relationship problems and prosocial behaviour.[70] | Parent and teacher | ● | ● | ● |
| Quality of life | **Child Health Utility 9D (CHU9D)**. 9-item measure of child quality of life.[62] | Parent | ● | ● | ● |
| School attendance | **School attendance** over the preceding 3 months. | Parent | ● | ● | ● |
| Academic functioning | **Wide Range Achievement Test 4 (WRAT)**. Spelling and math computation subtests to assess academic functioning.[71] | Child | | | ● |
| Academic functioning | Permission to access **NAPLAN** results (Years 3, 5, 7). NAPLAN is a national assessment of literacy and numeracy Australian children complete in Grades 3, 5, 7 and 9. | Linkage | | | |
| Cognitive functioning | **NIH Toolbox**. Cognitive Domain Tasks to assess cognitive functioning.[72] iPad administered assessment of executive function, attention, episodic/working memory, processing speed, language abilities, new learning and reading. | Child | | | ● |
| **PARENT OUTCOMES** | | | | | |
| Mental health | **Kessler 10 (K10)**. A 10-item validated measure of adult psychological distress.[73] | Parent | ● | ● | ● |
| Parenting stress | **Parenting Stress Index 4SF (PSI-4SF)**. 36-item measure of parenting stress.[74] Provides a measure of total parenting stress and three subscales reflecting the major sources of parenting stress (parental distress, difficult child, parent–child dysfunctional interaction). | Parent | ● | ● | ● |
| Quality of life | **Assessment of Quality of Life (AQoL4D)**. 12-item measure of parent quality of life.[75] | Parent | ● | ● | ● |
| **ECONOMIC OUTCOMES** | | | | | |
| Resource use | Family reported **service use** over preceding 3 months. | Parent | ● | ● | ● |
| Health service utilisation | Permission to access both **Medicare and Pharmaceutical Benefits Scheme (PBS)** data for 3 months prior to baseline up until the last follow-up assessment. | Linkage | ● | ● | ● |

●Measure used at indicated time point.
3, 3 months; 6, 6 months; ASD, autism spectrum disorder; B, baseline; NIH, National Institutes of Health.

Medication information is captured at baseline and 3- and 6-month follow-up. Based on prior work with children with ADHD and ASD we do not anticipate any risks, side-effects or discomfort to participants in completing our measures. In the event that a parent becomes distressed when completing questionnaires related to

stress or mental health, we have included information to link participants to a psychologist on our research team as well as directing parents to contact their general practitioner.

We have chosen the CSHQ as our primary outcome measure as this is a widely used and validated measure of behavioural sleep problems in children,[49] and was used in our pilot *Sleeping Sound* ADHD-ASD study.[48] It has been used to examine sleep behaviour in children with ASD and correlates highly with the objective measure of actigraphy.[57] It contains 33 items and includes 8 subscales: Bedtime Resistance (eg, *Child falls asleep in own bed*), Sleep-Onset Delay (eg, *Child falls asleep in 20 min after going to bed*), Sleep Duration (eg, *Child sleeps the right amount*), Sleep Anxiety (eg, *Child needs parent in the room to fall asleep*), Night Waking (eg, *Child moves to someone else's bed during the night*), Parasomnias (eg, *Child talks during sleep*), Sleep Disordered Breathing (eg, *Child snores loudly*) and Daytime Sleepiness (eg, *Child has difficulty getting out of bed in the morning*). A 3-point rating scale from 1=*rarely* to *3*=*usually* is applied to each item, indicating the sleep problem frequency during the past week. Some items are reverse coded and higher scores are indicative of more sleep problems. The measure has good internal consistency (α=0.78) and each subscale also has adequate internal consistency (α=0.93–0.56).[49]

Knowledge of participant group status will be limited to the project coordinator and study clinicians. If a family spontaneously reveals their intervention status to another team member, this is noted on the participant file. Given that it is impossible to maintain participant blinding in a trial of this nature, we have carefully chosen some outcome measures that can be assessed in a blinded fashion. For example, we will include a blinded teacher-reported measure of sleep, behavioural, social and motor functioning. We will not inform teachers of the child's group status and will ask parents not to inform teachers. We will also include blinded objective child measures of executive and school-based functioning.

## Sample size

In our pilot study,[48] we found a moderate to large effect of the intervention on sleep for total CSHQ score (standardised mean difference=0.7). However, since effect sizes from small studies may not be replicated in larger studies, we conservatively estimate a moderate standardised mean difference of 0.5 at 3-month and 6-month follow-up. Required sample size for this standardised mean difference was calculated using the equation $n=2(Z\alpha+Z[1-\beta])^2 \times SD^2/d^2$, where $Z_\alpha$ was set at p<0.05 (two tailed), $Z_{1-\beta}$ (power) was set at 0.80, d was set at 0.5 as per above and the SD for this standardised mean difference was set at 1. The resulting sample size estimate was then adjusted for the impacts of attrition (allowing for 20% loss to follow-up) and clustering (by individual and paediatrician). Clustering was handled with adjustments for design effect, assuming intraclass correlation (ICC)=0.08 for the within-child's paediatrician clustering (average cluster size=4, SD=3) and ICC=0.20 for the within-person clustering to control for repeated measurement due to control of baseline scores (estimated based on unpublished pilot data).[58] Allowing for 20% attrition over time and these forms of clustering, we will randomise a minimum of 234 children (117 in each group) to ensure sufficient power postintervention with our primary outcome time point at 3 months post-randomisation. Full calculations are available in the online supplementary file.

Strategies have been put in place to encourage participant retention and the completion of the follow-ups. These include a four-step reminder process for completing follow-up assessments and a study newsletter to promote family engagement.

## Data analysis

Primary analysis will be intention-to-treat (within participant data included as per initial treatment group allocation, regardless of actual intervention received) and will compare outcomes of children in the TAU and intervention groups at the 3- and 6-month post-randomisation time points, though our primary endpoint is 3 months post-randomisation. Analysis of the primary outcome (ie, child sleep problems) and all secondary outcomes (child—teacher reported daytime sleepiness, sleep hygiene, emotional and behavioural problems, social communicative symptoms, cognitive performance, academic achievement, school attendance and quality of life; parent—parenting stress, mental health, quality of life and work attendance) will be carried out using linear mixed effects regression with results presented as mean differences (and 95% CIs). For each outcome, a single mixed-effects model will be fitted incorporating baseline, 3-month and 6-month data using random effects to allow for the repeated measures within an individual, and clustering of individuals within the child's paediatrician. Separate parameters will be estimated for group differences at 3-month and 6-month postintervention; scores at baseline are added as a covariate. Differences in the outcomes of intervention and TAU groups will be assessed with linear mixed models, with individuals clustered within the child's paediatrician and assessment times clustered within individuals. We will run an unadjusted model and compare this to an adjusted model, with the unadjusted model as our primary analysis, and adjusted results to evaluate robustness of these findings. The adjusted model will include the following as covariates: child gender, age, ASD symptom severity (measured using the Social Communications Questionnaire—current form,[53] medication use (obtained from the Resource Use Questionnaire) and SES (measured according to the families residential postal code and using the index of relative socioeconomic disadvantage).[47 59] The child's paediatrician and individual participant will be modelled as random effects and the remaining variables will be handled as fixed effects. To account for missing data, the mixed models will be fitted using conditional maximum likelihood. As this approach makes an untestable assumption that data

are missing at random, we will perform sensitivity analyses in the form of pattern mixture models to explore the effect of departure from this assumption. A per-protocol analysis will also be undertaken as a supplementary analysis. Significance will be tested at p<0.05 for the primary analysis, but adjustments for type I error inflation will be made for secondary outcomes by using the false discovery rate method.[60]

The number of siblings in the trial is expected to be low based on our prior work. Even if the proposed mixed effects model were able to run with family as a clustering factor despite low number of sibling pairs, there is still some concern about obtained parameter estimates.[61] As such, alternative modelling approaches that can handle sparse data (such as generalised estimating equations) will be used as a sensitivity analysis to evaluate impact of clustering due to the presence of siblings on results obtained from our model without this correction.[61]

## Economic evaluation

An important secondary aim of the project will be to determine the cost-effectiveness of this intervention compared with TAU. The inclusion of the Child Health Utility 9D at baseline, 3-month and 6-month follow-up will allow calculation of QALYs as part of a cost–utility analysis, applying the area-under-the-curve approach.[62] A supplementary cost-effectiveness analysis will also be undertaken using the total CSHQ score as an outcome measure. The economic analysis will be conducted from a societal perspective, since interventions targeting children with ASD are likely to have benefits and costs beyond the narrow health sector (eg, ASD intervention services and productivity impacts for parents). A secondary analysis from a healthcare perspective will also be undertaken. The evaluation will first measure and value the use of healthcare resources over the period of the study between the two arms of the trial and then compare any additional costs to the additional outcomes achieved. Costs of the intervention will be measured using the financial records of the study team. Other costs incurred by children and their families will be measured using a Resource Use Questionnaire. Components of the Resource Use Questionnaire comprise the child's use of healthcare services, including: (1) inpatient services and emergency services, (2) health professional visits, (3) medication and (4) other resources used. The questionnaire also collects data on out-of-pocket and travel costs, as well as on a child's school attendance and parent's productivity. Participants will also be asked for their permission to access their MBS and PBS data for a time period of 3 months prior to baseline up until the last follow-up assessment. These data are not subject to recall bias and provide complete resource use costs (and out-of-pocket costs) for all MBS and PBS reimbursed healthcare services. Non-MBS and non-PBS funded services (eg, state-funded ASD services) will be captured in the Resource Use Questionnaire. Generalised linear models will be used to assess mean difference in costs

and outcomes between the two arms, adjusted for any differences between groups at baseline. An incremental cost-effectiveness ratio and an incremental cost–utility ratio will be calculated as the difference in average cost between the groups, divided by the difference in average CSHQ and QALYs, respectively. Non-parametric bootstrapping will be undertaken from the observed cost/QALY pairs (1000 simulated replications) to determine CIs and presented in a cost-effectiveness plane along with a cost-effectiveness acceptability curve. Sensitivity analyses will be carried out to explore the robustness of the results. While MBS and PBS data will be used for the base case analysis, a sensitivity analysis will be performed using self-report data derived from the Resource Use Questionnaire for those health services not captured by MBS and PBS.

## ETHICS AND DISSEMINATION

Behavioural sleep problems are a common burden associated with ASD. The proposed RCT of the *Sleeping Sound* intervention in ASD children with sleep problems will result in a significant advance in research on the treatment of sleep problems. The study is well-justified with reference to the demonstrated success of this tailored sleep intervention in ADHD,[47] our strong pilot data,[48] and the potential advantages of the intervention in alleviating social, emotional and academic problems connected to the sleep disorder in the children themselves and their parents. If the intervention is found to be effective in this sample of children who are of borderline/average intellectual ability, we will subsequently be able to explore whether the intervention is successful for children with ASD who also have an intellectual disability. Treatment recommendations for treating sleep problems in children with ASD have been released.[63] The Sleep Committee of the Autism Treatment Network recommend the use of behavioural intervention as a first-line therapy.[64] The Autism UK National Institute for Health and Care Excellence clinical guidelines (No 170) for sleep management recommend development of individualised behavioural intervention which includes specialist monitoring and use of a sleep plan as a central concept of sleep management.[65] To date, the treatment of sleep disturbance in children with ASD is based on expert opinion and consensus practise, informed by limited evidence. Results from this study will inform evidence-based clinical practice and guide appropriate selection and tailoring of behavioural strategies to address specific sleep problems. We aim to publish our findings in peer-reviewed journals and present our research at international conferences and community organisations. The study has been granted ethical approval by the Royal Children's Hospital (No 36154), Deakin University (No 2017–130), the Victorian Department of Education and Early Childhood Development (No 2016_003134) and the Catholic Education Office (Melbourne No 0501) Human Research Ethics Committees. Important protocol changes are reported to each ethics committee via ethics amendments.

## DATA MANAGEMENT

All participant data will be confidential and hardcopy data will be securely stored in locked offices for which only research team members have access to the keys. All online participant data are collected and stored in REDCap,[54] which is a secure database stored on the Deakin University network, accessible only to the research team via individual usernames and passwords. All chief investigators will have access to the final dataset. Information will be kept for 7 years after the youngest participant's 18th birthday or 15 years after the completion of the study, whichever date is later. After this time, the data will be destroyed.

Data from all hardcopy surveys will be entered into REDCap by a member of the research team.[54] To ensure quality data, a process of data checking and cleaning will be undertaken by the research team. We will keep a register of data issues to ensure problems with data are addressed in a timely manner and conduct regular audits of data management processes.

**Author affiliations**
[1]Deakin University, Geelong, Victoria, Australia
[2]Murdoch Children's Research Institute, Parkville, Victoria, Australia
[3]Department of Paediatrics, The University of Melbourne, Melbourne, Victoria, Australia
[4]The Royal Children's Hospital, Parkville, Victoria, Australia
[5]Brain and Mind Centre, University of Sydney, Camperdown, New South Wales, Australia
[6]Institute of Psychiatry, Psychology and Neuroscience, King's College London, London, UK

**Contributors** NR, NP, ES, HH, KW, JM, PH and CM conceived the study. STB, DM and LE contributed to the writing of the methods and intervention for the study. NP and STB drafted the manuscript. MF-T contributed to the design and analytical components of the study. All authors contributed, read and approved the final manuscript.

**Funding** This work was supported by the Australian National Health and Medical Research Council (NHMRC; project grant no APP1101989). ES is funded by an NHMRC Career Development Fellowship 1110688 (2016-21) and a veski Inspiring Women Fellowship. HH is supported by an NHMRC Practitioner fellowship 1136222 (2018-2022). The Murdoch Children's Research Institute is supported by the Victorian Government's Operational Infrastructure Support Program.

**Competing interests** NP, NR and JM receive philanthropic funding from the Moose Foundation, Ferrero Group Australia as part of its Kinder + Sport pillar of Corporate Social Responsibility initiatives, MECCA Brands, Wenig Family, Geelong Community Foundation, and Grace & Emilio Foundation; and industry partner funding from the Victorian Department of Education, to conduct research in the field of neurodevelopmental disorders and inclusion. NP, NR and JM have also previously received funding from the Australian Foortball League and industry partner funding from the NDIS. NR has received donations form Vic Health and Bus Assoication Victoria; and previously received speaker honorarium from Novartis (2002), Pfzier (2006) and Nutricia (2007); and is a Director of the Amaze Board (Autism Victoria). None of the companies or organisational bodies listed above had a role in this research including the collection, analysis and interpretation of data; in writing of the manuscript; and /or in the decision to submit the article for publication.

**Patient consent for publication** Not required.

**Provenance and peer review** Not commissioned; externally peer reviewed.

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
