## [Reviewer comments · BMJ Open]

ARTICLE DETAILS

TITLE (PROVISIONAL)	Sleeping Sound with Autism Spectrum Disorder (ASD): Study protocol for an efficacy randomised control trial of a tailored brief behavioural sleep intervention for autism spectrum disorder.
AUTHORS	Papadopoulos, Nicole; Sciberras, Emma; Hiscock, Harriet; Williams, Katrina; McGillivray, Jane; Mihalopoulos, Cathrine; Engel, Lidia; Fuller-Tyszkiewicz, Matthew; Bellows, Susannah; Marks, Deborah; Howlin, Patricia; Rinehart, Nicole

VERSION 1 – REVIEW

REVIEWER	Sela Sanberg, Ph.D., BCBA-D Western Michigan University, U.S.A.
REVIEW RETURNED	28-Feb-2019

GENERAL COMMENTS	I would like to commend the authors on their aims to evaluate the effectiveness of a brief behavioral treatment on the reduction of childhood sleep problems in children with Autism Spectrum Disorder (ASD) and further our understanding of the relationship between childhood sleep disturbances and the severity of behavioral issues associated with ASD. However, there are concerns related to the currently described methods, measurement of sleep variables, and APA formatting. Therefore, my recommendation is for major revisions to prepare the manuscript for publication. Below are comments for the authors' consideration, should there be an interest in revising the current protocol. Comments First, this topic has a body of literature that was not adequately referenced in the manuscript. For example, the integration of existing literature on correlates of sleep disturbances and autism symptom severity (Fadini et al., 2015; Richdale & Baglin, 2015; Tudor et al., 2012) is needed. The introduction may also benefit from a stronger argument in regards to the impact on parents (Hodge et al., 2014; Levin & Scher, 2016; Mihaila & Hartley, 2016) and quality of life (Tilford et al., 2015). Please see Vriend et al., (2011) and Bereford et al., (2018) for reviews of behavioral and pharmacological interventions for sleep disturbances and children with ASD. Second, the method section does not adequately describe the steps necessary to replicate the study. For example, all participants are reported to have "a paediatrician confirmed multidisciplinary diagnosis of ASD", without a criterion for establishing an ASD diagnosis. The gold standard and most reliable tools available for diagnosing autism (the Autism
---

	Diagnostic Observation Schedule, 2nd ed. (ADOS-2)) and rating severity (the Autism Diagnostic Interview-Revised (ADI-R) and the Childhood Autism Rating Scale, 2nd ed, (CARS-2)), were not reportedly used in the study. How will this be addressed? As for the children currently using medication, how will possible changes in medication use be controlled for? This was not adequately addressed in the reported limitations of the study. For the dependent measures, a more comprehensive description of each tool is needed, as well as how each measure will be used to inform outcomes. For example, the use of a sleep diary is referenced without any indication as to which sleep variables are being recorded. This also leaves the reader wondering: Are parents trained in recording sleep variables on a sleep diary? How are the parents trained to accurately record data? How will over or under reporting be addressed? Additionally, actigraphy was mentioned in reference to CSHQ validity, only. Will an objective sleep measure be used? If not, how will subjectivity be addressed in the limitations? More importantly, the sleep variables noted lack clarity and definition. For example, on page 15, sentence 52, "...prolonged nighttime waking" are reportedly treated with a bedtime fading procedure. However, it is not clear if the authors intend to treat frequent night awakenings (FNA) or waking after sleep onset (WASO) or some other behavioral issue. Therefore, the use of established terminology for reporting sleep variables is needed. It is also recommended that the authors adhere to the APA Publication Manual standards in revised submissions. Errors in spacing, grammar, and citations will need to be appropriately addressed.
--	--

REVIEWER	Emily A. Abel Yale University, USA
REVIEW RETURNED	05-Mar-2019

GENERAL COMMENTS	Thank you for the opportunity to review the protocol titled "Sleeping Sound with Autism Spectrum Disorder: Study Protocol for an Efficacy RCT of a Tailored Brief Behavioral Sleep Intervention for Autism Spectrum Disorder." This protocol outlines an important study—with strong implications for improving quality of life for both children with ASD and their families. Overall, the protocol is well-written and addresses a critical gap in the literature. However, there are some areas where clarification, expansion, or re-organization would improve the quality of the protocol for publication. My comments (most of which are to improve clarity), are outlined below. Introduction: 1. Rather than saying 'resistance going to bed' with bedtime resistance in parentheses, it would be more succinct to just state bedtime resistance the first time, especially since this is the term used throughout the rest of the protocol and resistance going to bed doesn't add to defining the term.
---

	2. The sentence: “Furthermore, sleep problems in ASD have been shown to be more persistent compared to typically developing children” is a bit unclear and seems tacked on in its current location. Do you mean that sleep problems persist across development in children with ASD as opposed to TD? Or that rates are higher in ASD than TD children. I recommend clarifying the sentence and would consider moving this sentence to come after the first sentence in that paragraph where you describe rates of sleep problems in this population. 3. I recommend checking the percentages you are citing for rates of sleep problems in ASD. Generally, it is estimated that 40-80 or 50-80% of these children have elevated sleep problems. There are definitely papers reporting samples with less than 73%, so we should be careful to be as accurate as possible when reporting these estimates that will be later cited. 4. Since this study actually excludes individuals with intellectual disability, the last sentence on page 6 really isn’t needed and leads the reader to believe this is a construct that will be measured in your efficacy trial although it is not. 5. On page 7, you mention that sleep problems are associated with psychopathology, but most of the examples/studies that come after this (i.e., attention) are not really psychopathologies. You may want to choose another term for this or state that sleep is associated with behavioral correlates and psychopathology (or something to that effect). 6. I might consider re-framing the first portion of the paragraph on treating sleep problems in children with ASD. There are many pharmacological interventions recommended to children with ASD to treat sleep (in addition to melatonin). I suggest setting this up to convey “behavioral sleep interventions are recommended as the first line of treatment for childhood sleep problems” or something to that effect, rather than even mentioning that melatonin is often prescribed. You could add –even before medication if you think that remains an important point to make. However, I would be hesitant, because although behavioral sleep interventions should be the first line of intervention, there are some sleep disorders that are best treated with pharmacological interventions and we should be careful not to send the message to families that these are inherently bad. Aims and Hypotheses 1. When describing the study aims, I recommend saying whether the brief sleep
--	---

	intervention delivered by study conditions is related to the following outcomes or to determine how the sleeping sound intervention relates to the following outcomes. 2. I would consider condensing your aims and hypotheses as the list is rather burdensome. Your hypotheses are already somewhat incorporated in your list of outcomes (associated with the aims of the study). For example, you list reductions in child sleep problems, and improvements in child and primary caregiver functioning, which is already what you hypothesize. Aside from a few added pieces which you could incorporate, the list of hypotheses seems redundant. Methods 1. It may make sense to move the sample size paragraph to come either directly after your description of participants, or after the section on measures that leads into data analysis. I would suggest the latter, if you decide to move it. 2. The description of participants, recruitment, and randomization, and the intervention itself are clear. However, I think the protocol would benefit from more information about the timeline and procedure. For example, walking us through baseline, intervention, and follow-up procedures. What happens at each of these. This could be added in a procedure section or put added to another logical place within the protocol. You may also consider some kind of timeline graphic that describes what happens at each phase of the study (i.e., what measures are given). 3. I recommend providing additional information about the CSHQ in-text since this is the primary measure you are using to assess change in sleep. For example, what components will you be looking at specifically, describe the subscale, scoring, etc. You do reference a table for all measures, which is great (and I think appropriate for the remainder of the measures). I suggest, if possible, making the table longer so (i.e., landscape) so the outcome can fit on one line. I would also suggest bolding the name of the measure and using shading to separate each measure's info from the next line. For example, you could alternate grey and white shading for every other measure. Other minor comments: Although the paper is well-written overall, please carefully check the paper for typos, grammatical errors (i.e., verb-tense agreement), and wording of some sentences. There are still errors throughout. See below for a few examples.  • “Further, it is unclear whether behavioural sleep interventions for children with ASD leads to improvements...” This should be “lead to improvements”.
--	--

	 • “More recently, Hiscock et al. 2015 has shown” should be “showed” or “demonstrated” or something similar • On page 10, the last sentence in the paragraph above aims and hypotheses could be reworded. For example, it may sound better to say: “Given that our pilot findings showed...” and “can improve” rather than improves. My last comment is whether this project is registered as a clinical trial/RCT? I may have missed it, but I didn't see this information anywhere. It should be provided if it is registered. If not, disregard. Thank you again for the opportunity to review this protocol. I thoroughly enjoyed reading it and look forward to the possibility of seeing it published, along with empirical papers using data from this study
--	---

REVIEWER	Erik Cobo Valeri Barcelona-Tech (UPC)
REVIEW RETURNED	05-Apr-2019

GENERAL COMMENTS	I read the protocol document BMJ-open-2019-029767 about a behavioural sleep intervention for autistic spectrum disorder. I think it is a well written document, presented before the end of the recruitment that may be worth the publication in BMJ-open. Unfortunately, I do not believe that this document will allow future researchers to reproduce your methods. You can improve the reproducibility of the methods by following the recommendations in the reporting guidelines. In fact, BMJ-open instructions to authors say: "Please use these guidelines to structure your article. Completed applicable checklists, structured abstracts and flow diagrams should be uploaded with your submission; these will be published alongside the final version of your paper." So, please follow and provide the SPIRIT checklist, specifying where any element is reported. For example, you must specify the masked status for any researcher, not just for the outcome assessors. Your sample size rationale was not completely clear to me. Please add the spirit figure, clarifying that recruitment precedes randomization. Keep in mind that your final report should also be reported according to the CONSORT extension for non-pharmacological interventions. So, please, address now any methodological requirements in this extension. For example, specify the selection criteria and the training of the interventionists. As masked evaluators could be spontaneously un-blinded, please fully specify any actions to preserve blinding. Please also address the methodological elements of the CONSORT extension to cluster trials. You increased the sample size to "allow" a 20% attrition, but the main analysis is by ITT. To avoid the risk of bias due to attrition, consider useful insights to prevent and treat missing information at http://www.nejm.org/doi/full/10.1056/NEJMSr1203730 I liked your statistical analysis, although it is specified in a vague way, which may open the risk of bias due to selective outcome reporting. To fully specify the analysis avoiding this risk, consider also adhering to the recommendations at https://jamanetwork.com/journals/jama/fullarticle/2666509.
--

	As your trial has already started, the goal of following all these recommendations is not to change your design, but to be completely transparent and clear about your methods. Please consider asking the journal for additional time to update the manuscript.
--	--

REVIEWER	Baptiste Leurent London School of Hygiene and Tropical Medicine
REVIEW RETURNED	09-Apr-2019

GENERAL COMMENTS	Overall this appears a well-designed trial and the protocol would, in my view, be relevant for BMJ Open. I have only “minor” comments, requiring clarifications. My main comment is about the sample size, which is currently difficult to follow. I commented mainly on the statistics/economic evaluation, but the rest of the manuscript looked fine. Abstract (page 4): The sample size should be in future tense or “aims to recruit 234 children”. Sample size (page13-14): I found the sample size calculations difficult to follow:  - “effect size” of 0.7, I suspect you mean a “standardised effect size”. You need to be clear that this is not the difference on the CSHQ scale, but divided by the standard deviation. What would it correspond to on the CSHQ scale? - Please define ICC - You mention an ICC for the recruitment centre. It is not clear why this would matter, given that it is an individually-randomised trial. Where does this value of 0.08 come from? - Does SD=3 refer to the SD of the cluster size? (It sounds large if the average is 4) - You mention an ICC for within-person clustering, but again, why does it matter given your primary analysis is based on a single follow-up measure? Or is it to take into account of your correlation between baseline and follow-up? Where does 0.20 come from? - “Results showed that we will need to analyse results of 188 individuals”. Where do this result come from? If it is based on a formula or a software, indicate which. Note that the final sample size (n=234) looks reasonable in my opinion, and I would not change it at this stage, but the formula you used should be more transparent. If it was based on the simulations, please provide the code as supplementary material. P14, line 36: which range for the blocks size? P14, l42 “girls, boys, and other genders”: did you really stratify gender in 3 categories? I would suspect too few participants in the “other genders”. P14, l50 : “all subsequent sibling allocated to same treatment group”: This dependent allocation affects your analysis/sample size, but suppose will be a minority? (otherwise discuss) P17:
---

	The list of collected measures (Table2) is clear, but I would expect to see a list of which outcomes will be analysed as primary and secondary outcomes precisely. Perhaps add a paragraph between “Measure” and “Data analysis”. Or will all measures collected be compared between groups? If so, please state this explicitly, and list all these outcomes in the text. Also indicate at which time-point they will be analysed. If relevant, specify which are more “key secondary outcomes” and which are more “exploratory” (see comment on multiplicity below). Data analysis (page17-18): Line 40: Clarify “intent-to-treat” as it can have different interpretations. (regardless of the intervention effectively received?) Line 40: clarify the secondary analyses at 3M and 6M. Will both time-point be analysed separately? Or in a mixed-model (assuming different effect at 3 and 6M)? Also for the primary analysis, will the 3M outcome be analysed alone, or in a mixed effect with baseline and 3M data, or with baseline, 3M and 6M data? Page17, line 59: The baseline covariates should ideally be defined a priori. If a factor is likely to be an important predictor of CSHQ, I would probably adjust for it in all cases. It is fine to adjust for imbalance observed a posteriori, but probably more as a sensitivity analysis. You should also be clearer about these baseline factors. For example, how socio-economic status, parent education, mental health and sleep medication will be defined exactly? Page 18, Line 6, is it the centre or the paediatrician that will be modelled as random-effect? Or are they exactly the same? It seems the trial includes numerous outcomes, which will be compared at both 3M and 6M. Are you going to report p-values for all of these outcomes? Are you going to try to adjust for multiple testing? Is there some outcomes/time-points more important than others? Page18-19, economic evaluation: Line 31: Will all the 15 outcomes (in table 2) really be compared for the cost-consequences analysis? At which time point(s)? Line 47: Clarify “any change”. Does it mean the “difference” in resource use between baseline and the follow-up period will be analysed (how?). Line50: How will the costs and outcomes be compared between arms exactly (same model than for the outcome data analysis? Will it be adjusted?). Page19, Line17: Will Medicare and PBS collect data similar to the resource-use questionnaire? If so, which one will be used/how both information will be combined.
--	---

REVIEWER	Kim Madden McMaster University, Canada
REVIEW RETURNED	22-Apr-2019

GENERAL COMMENTS	Thank you for the opportunity to review this protocol for an RCT investigating a "Sleeping Sound" intervention for children with ASD. The topic is interesting and relevant to readers. The methodology is also very good but I have a few recommendations and questions for clarity that I hope the investigators will consider.  1. Under aims and hypotheses, there are sections titled "Primary outcome" and "secondary outcomes" which are phrased like hypotheses. The authors should not conflate objectives/hypotheses with outcomes. These should be two separate sections. 2. I recommend clearly stating in the methods section whether the protocol conforms to SPIRIT reporting requirements. 3. The authors report that the study started in 2016. I suggest explicitly stating the stage the trial is at, at the time of submission. E.g. is recruitment ongoing or completed? 4. Under the "patient involvement" section: This section only refers to satisfaction with the intervention itself but not necessarily with the trial. Were patients/caregivers involved in designing the trial, e.g. selecting the most important research questions and/or outcomes or determining a minimally important difference (MID) for selected outcomes? 5. Will children be asked for assent? if not, how will the investigators ensure that the children wish to participate (even if they cannot give full legal consent)? 6. In the sample size section: Do the investigators have an estimate of the minimally important difference (MID) of their primary outcome? Is this sample size large enough to detect the MID? 7. In the randomisation section: It looks like allocation concealment will not be an issue, but I recommend commenting explicitly on allocation concealment. 8. The randomisation section states that siblings will be allocated to the same intervention. What percentage of families are expected to enroll multiple children? This becomes a sort of cluster trial where the unit of randomization is a family. I recommend explicitly stating the unit of randomization in this paper. It may have an impact on the statistical methodology required depending on how many children are enrolled this way. If a single family enrolls two children, does this count as two participants toward enrollment or one? This will affect sample size and needs to be clarified. 9. How will co-interventions be minimized/dealt with? It would be unlikely that TAU participants would fully cross-over into the treatment group but they may be trying similar interventions on their own, or supplements, medications etc that may have some effect on sleep. 10. The SPIRIT statement requires an explanation of data collection/management processes, quality assurance/monitoring, a description of how participant retention will be maximized/addressed, any safety monitoring, interim analyses planned, and whether/how harms will be addressed. I recommend that the authors add these descriptions and double-check that all SPIRIT statement items are explicitly addressed in the protocol (spirit-statement.org). 11. Only one limitation is mentioned in the point form strengths and limitations section. The authors should discuss whether there
--

	are likely to be any more limitations and how they will be/have been addressed either by design or analysis.
--	--

VERSION 1 – AUTHOR RESPONSE

Reviewer 1	
Reviewer comments	Authors response
4. This topic has a body of literature that was not adequately referenced in the manuscript. For example, the integration of existing literature on correlates of sleep disturbances and autism symptom severity (Fadini et al., 2015; Richdale & Baglin, 2015; Tudor et al., 2012) is needed. The introduction may also benefit from a stronger argument in regards to the impact on parents (Hodge et al., 2014; Levin & Scher, 2016; Mihaila & Hartley, 2016) and quality of life (Tilford et al., 2015). Please see Vriend et al., (2011) and Bereford et al., (2018) for reviews of behavioral and pharmacological interventions for sleep disturbances and children with ASD.	Thank-you for your suggestions. The section of the introduction relating to impacts on behaviour and psychopathology has been expanded (pg 5). The section of the introduction relating to impacts on the family has been expanded. Levin & Scher, 2016 has not been included as the article relates to toddlers rather than children; Mihaila & Hartley, 2018 has not been included as it relates to parent sleep quality as opposed to child sleep quality (pg 5). Thank you – we have now included these references (pg 5, 6 and 11).
5. The method section does not adequately describe the steps necessary to replicate the study. For example, all participants are reported to have “a paediatrician confirmed multidisciplinary diagnosis of ASD”, without a criterion for establishing an ASD diagnosis. The gold standard and most reliable tools available for diagnosing autism (the Autism Diagnostic Observation Schedule, 2nd ed. (ADOS-2)) and rating severity (the Autism Diagnostic Interview-Revised (ADI-R) and the Childhood Autism Rating Scale, 2nd ed, (CARS-2)), were not reportedly used in the study. How will this be addressed?	We have designed our intervention so that it can be feasibly delivered within the Australian healthcare scheme and in the context of typical paediatric and psychological consultation durations. The requirements for an ASD diagnosis used in the study are in line with current Australian diagnostic guidelines involving a multidisciplinary and comprehensive clinical assessment by a multidisciplinary team. This has been included on pg. 10 of the revised manuscript standard autism diagnosing tools such as the ADI-R and ADOS-2 in this trial given the already extensive time commitment requested of participants. In

		addition to obtaining a multidisciplinary diagnosis of ASD, participants were
	As for the children currently using medication, how will possible changes in medication use be controlled for? This was not adequately addressed in the reported limitations of the study.	also administered the Social Communication Questionnaire- Lifetime form to screen for severity of autism symptoms are were required to meet a cut-off score of ≥ 11 on this measure to be eligible for this study We are collecting medication information (typ baseline, 3, 6 and 12 months to track any change Medication information is also captured with the P benefits scheme information. Medication use will a covariate (please refer to the data analysis sec 18).
6.	For the dependent measures, a more comprehensive description of each tool is needed, as well as how each measure will be used to inform outcomes. For example, the use of a sleep diary is referenced without any indication as to which sleep variables are being recorded. This also leaves the reader wondering: Are parents trained in recording sleep variables on a sleep diary? How are the parents trained to accurately record data? How will over or under reporting be addressed? Additionally, actigraphy was mentioned in reference to CSHQ validity, only. Will an objective sleep measure be used? If not, how will subjectivity be addressed in the limitations?	Additional details have been added to the table for our measures (pp. 2425). The sleep diary is not a sleep dependent measure, but one component of the sleep intervention. Parents are instructed how to complete this at the time of intervention We are using published validated dependent measures which have established validity and reliability. No objective measures of sleep are used. Correlation of actigraphy with the CHSQ indicates that it provides a valid measure of sleep patterns similar to those of an actigraphy device. We carefully considered the use of actigraphy in this RCT but chose not to for several reasons. In the ADHD Sleeping Sound trial, we attempted to use actigraphy but were only able to do so for 66/244 children. Children often destroyed, discarded or refused to wear the actiwatch, as has been reported in other large-scale RCTs of children with developmental disorders including ASD (e.g. Gringras et al. (2012). Inclusion of actigraphy would also increase participant time and trial costs.
7.	More importantly, the sleep variables noted lack clarity and definition. For example, on page 15, sentence 52, "...prolonged nighttime waking" are reportedly treated with a bedtime fading procedure. However, it is not clear if the authors intend to treat frequent night awakenings (FNA) or waking after sleep onset (WASO) or some other behavioral issue. Therefore, the use of established terminology for reporting sleep variables is needed.	We have added more detail and consistent language to address this in the manuscript (pg. 13).

8.	It is also recommended that the authors adhere to the APA Publication Manual standards in revised submissions. Errors in spacing, grammar, and citations will need to be appropriately addressed.	The manuscript has been updated to comply with APA standards, except for citations and referencing as the journal uses a modified vancouver approach.
----	---	---

Reviewer 2		Authors response
Introduction		
9.	Rather than saying 'resistance going to bed' with bedtime resistance in parentheses, it would be more succinct to just state bedtime resistance the first time, especially since this is the term used throughout the rest of the protocol and resistance going to bed doesn't add to defining the term.	This has been updated as suggested (pg 4).
10.	The sentence: "Furthermore, sleep problems in ASD have been shown to be more persistent compared to typically developing children" is a bit unclear and seems tacked on in its current location. Do you mean that sleep problems persist across development in children with ASD as opposed to TD? Or that rates are higher in ASD than TD children. I recommend clarifying the sentence and would consider moving this sentence to come after the first sentence in that paragraph where you describe rates of sleep problems in this population.	This sentence has now been clarified (pg 4). Due to other changes made to the manuscript the sentence has been left in its current location.
11.	I recommend checking the percentages you are citing for rates of sleep problems in ASD. Generally, it is estimated that 40-80 or 50-80% of these children have elevated sleep problems. There are definitely papers reporting samples with less than 73%, so we should be careful to be as accurate as possible when reporting these estimates that will be later cited.	Thank you for highlighting this. The percentages have now been updated (pg 2 and pg 4).
12.	Since this study actually excludes individuals with intellectual disability, the last sentence on page 6 really isn't needed and leads the reader to believe this is a construct that will be measured in your efficacy trial although it is not.	Thank you for raising this point. This sentence has now been removed as suggested.
13.	On page 7, you mention that sleep problems are associated with psychopathology, but most of the examples/studies that come after this (i.e., attention) are not really psychopathologies. You may want to choose another term for this or state that sleep is associated with behavioral	This has been updated as suggested (pg 5).

	correlates and psychopathology (or something to that effect).	
14.	I might consider re-framing the first portion of the paragraph on treating sleep problems in children with ASD. There are many pharmacological interventions recommended to children with ASD to treat sleep (in addition to melatonin). I suggest setting this up to convey “behavioral sleep interventions are recommended as the first line of treatment for childhood sleep problems” or something to that effect, rather than even mentioning that melatonin is often prescribed. You could add –even before medication if you think that remains an important point to make. However, I would be hesitant, because although behavioral sleep interventions should be the first line of intervention, there are some sleep disorders that are best treated with pharmacological interventions and we should be careful not to send the message to families that these are inherently bad.	Thank you. We have reframed the first portion of the paragraph to reflect these suggestions (pg 5).
Aims & Hypotheses		

15.	When describing the study aims, I recommend saying whether the brief sleep intervention delivered by study conditions is related to the following outcomes or to determine how the sleeping sound intervention relates to the following outcomes.	This has been updated as suggested (pg 8).
16.	I would consider condensing your aims and hypotheses as the list is rather burdensome. Your hypotheses are already somewhat incorporated in your list of outcomes (associated with the aims of the study). For example, you list reductions in child sleep problems, and improvements in child and primary caregiver functioning, which is already what you hypothesize. Aside from a few added pieces which you could incorporate, the list of hypotheses seems redundant.	Please see response to feedback item #41 below.
Methods		
17.	It may make sense to move the sample size paragraph to come either directly after your description of participants, or after the section on measures that leads into data	This has been moved to after the measures section as suggested.

	analysis. I would suggest the latter, if you decide to move it.	
18.	The description of participants, recruitment, and randomization, and the intervention itself are clear. However, I think the protocol would benefit from more information about the timeline and procedure. For example, walking us through baseline, intervention, and follow-up procedures. What happens at each of these. This could be added in a procedure section or put added to another logical place within the protocol. You may also consider some kind of timeline graphic that describes what happens at each phase of the study (i.e., what measures are given).	We have included a SPIRIT figure and Consort figure to address this in addition to our table of measures (See figure 1 and figure 2).
19.	I recommend providing additional information about the CSHQ in-text since this is the primary measure you are using to assess change in sleep. For example, what components will you be looking at specifically, describe the subscale, scoring, etc. You do reference a table for all measures, which is great (and I think appropriate for the remainder of the measures). I suggest, if possible, making the table longer so (i.e., landscape) so the outcome can fit on one line. I would also suggest bolding the name of the measure and using shading to separate each measure's info from the next line. For example, you could alternate grey and white shading for every other measure.	Additional information relating to the CSHQ has been provided in text as suggested (pg 15-16). The table has been updated as suggested (pp 24-25).
Other Minor Comments		
20.	Although the paper is well-written overall, please carefully check the paper for typos, grammatical errors (i.e., verb-tense agreement), and wording of some sentences. There are still errors throughout. See below for a few examples.  • “Further, it is unclear whether behavioural sleep interventions for children with ASD leads to improvements...” This should be “lead to improvements”. • “More recently, Hiscock et al. 2015 has shown” should be “showed” or “demonstrated” or something similar 	Thank you. We have addressed all typos and grammatical errors in the manuscript.
	 • On page 10, the last sentence in the paragraph above aims and hypotheses could be reworded. For example, it may sound better to say: “Given that our pilot 	

	findings showed..." and "can improve" rather than improves.	
21.	My last comment is whether this project is registered as a clinical trial/RCT? I may have missed it, but I didn't see this information anywhere. It should be provided if it is registered. If not, disregard.	Yes, this RCT is registered. Registration details are included on Page 3 of the manuscript.
Reviewer 3		
	Reviewer comments	Authors response
22.	I do not believe that this document will allow future researchers to reproduce your methods. You can improve the reproducibility of the methods by following the recommendations in the reporting guidelines. In fact, BMJ-open instructions to authors say: "Please use these guidelines to structure your article. Completed applicable checklists, structured abstracts and flow diagrams should be uploaded with your submission; these will be published alongside the final version of your paper." So, please follow and provide the SPIRIT checklist, specifying where any element is reported. For example, you must specify the masked status for any researcher, not just for the outcome assessors. Your sample size rationale was not completely clear to me. Please add the spirit figure, clarifying that recruitment precedes randomization. Keep in mind that your final report should also be reported according to the CONSORT extension for non-pharmacological interventions. So, please, address now any methodological requirements in this extension. For example, specify the selection criteria and the training of the interventionists. As masked evaluators could be spontaneously un-blinded, please fully specify any actions to preserve blinding.	Thank you for this comment. We have now provided the SPIRIT checklist. The sample size section of the manuscript has been revised (pp 16-17). In particular, we elaborate on how the final sample size was arrived at, and that clustering effects were based on prior, preliminary work. A spirit figure has been provided. The order of events information is clearly stated in the Flow of Participants figure. The consort extension for non-pharmacological interventions has now been complied with, with updated made through the revised document.
23.	Please also address the methodological elements of the CONSORT extension to cluster trials.	As randomisation occurs at the participant level this trial is not a cluster trial.

24.	You increased the sample size to "allow" a 20% attrition, but the main analysis is by ITT. To avoid the risk of bias due to attrition, consider useful insights to prevent and treat missing information at http://www.nejm.org/doi/full/10.1056/NEJMsr1203730	Thank you for this useful resource. Our intended way of dealing with missing data is full-information maximum likelihood estimation. As such, we wanted to ensure the power analysis allowed for some parameters in the model (i.e., later assessments) to have less cases than others (baseline data). We do take on-board your suggestion though to consider the assumptions for this approach for handling missing data, and will undertake sensitivity analyses to evaluate the plausibility of these assumptions of missing at
-----	---	--

		random (for FIML), that cannot be directly tested but its effects may be evaluated through methods such as pattern mixture models. In the original submission, we had mentioned that we would undertake these sensitivity analyses, but did not provide sufficient information here to give a clearer indication of how this might be achieved. We have revised accordingly to make this clearer (pp 17-18).
--	--	--

25.	I liked your statistical analysis, although it is specified in a vague way, which may open the risk of bias due to selective outcome reporting. To fully specify the analysis avoiding this risk, consider also adhering to the recommendations at https://jamanetwork.com/journals/jama/fullarticle/2666509 .	We have now included extra information in the manuscript to justify our statistical analysis to avoid the risk of bias due to selective outcome reporting (pp. 17-18).
-----	--	--

Reviewer 4

Abstract

26.	The sample size should be in future tense or "aims to recruit 234 children".	This has been updated as requested (pg 2).
-----	--	--

Sample Size

27.	I found the sample size calculations difficult to follow:  - “effect size” of 0.7, I suspect you mean a “standardised effect size”. You need to be clear that this is not the difference on the CSHQ scale, but divided by the standard deviation. What would it correspond to on the CSHQ scale? - Please define ICC - You mention an ICC for the recruitment centre. It is not clear why this would matter, given that it is an individually-randomised trial. Where does this value of 0.08 come from? - Does SD=3 refer to the SD of the cluster size? (It sounds large if the average is 4) - You mention an ICC for within-person clustering, but again, why does it matter given your primary analysis is based on a single follow-up measure? Or is it to take into account of your correlation between baseline and follow-up? Where does 0.20 come from? - “Results showed that we will need to analyse results of 188 individuals”. Where do this result come from? If it is based on a formula or a software, indicate which. Note that the final sample size (n=234) looks reasonable in my opinion, and I would not change it at this stage, but the formula you used should be more transparent. If it was based on the simulations, please provide the code as supplementary material. 	Sorry for the confusion. We have clarified in the revised manuscript (pg. 1617):  1. The effect size is a standardized mean difference, which we then contextualise in terms of improvement in CSHQ scores for a more practical understanding of the impact of the intervention. 2. We have now defined ICC in its first use. 3. We have included a clustering effect at the level of child’s paediatrician based on pilot data showing this level of clustering (.08). We felt it was important to include in the calculations since individuals recruited from the same paediatrician may have similar experiences that lead to violation of the independence of errors assumption of standard regression models. 4. The within-person clustering estimate (.20) is also based on pilot data. 5. G*Power was used to calculate the sample size needed to detect a standardized mean difference of .5. The impact of attrition and clustering were then taken into account – clustering by calculating the design effect and subsequent effective sample size, and attrition by anticipating 20% drop off by follow-up.
28.	P14, line 36: which range for the blocks size?	We have now added information as to the block sizes (pg 12).
29.	P14, l42 “girls, boys, and other genders”: did you really stratify gender in 3 categories? I would suspect too few participants in the “other genders”.	Yes, we did stratify gender into three categories. We agree that it is very likely that this category will have too few participants to stand alone in analyses.
30.	P14, l50: “all subsequent sibling allocated to same treatment group”: This dependent allocation affects your analysis/sample size, but suppose will be a minority? (otherwise discuss)	We have now updated the data analysis section (pg 18).
Data Analysis		
31.	Line 40: Clarify “intent-to-treat” as it can have different interpretations. (regardless of the intervention effectively received?)	We have added to this sentence to make this clearer (pp 17-18).
32.	Line 40: clarify the secondary analyses at 3M and 6M. Will both time-point be	

	analysed separately? Or in a mixed-model (assuming different effect at 3 and 6M)?	We have now clarified this in the document. Will use a single model for each outcome (pp 17-18).
33.	Also for the primary analysis, will the 3M outcome be analysed alone, or in a mixed effect with baseline and 3M data, or with baseline, 3M and 6M data?	
34.	Page17, line 59: The baseline covariates should ideally be defined a priori. If a factor is likely to be an important predictor of CSHQ, I would probably adjust for it in all cases. It is fine to adjust for imbalance observed a posteriori, but probably more as a sensitivity analysis. You should also be clearer about these baseline factors. For example, how socio-economic status, parent education, mental health and sleep medication will be defined exactly?	We will run the model twice; once with adjustment for proposed covariates, and once without this adjustment. Both will be reported to demonstrate any differences in results. This has been reflected on pp 17-18 of the revised manuscript.
35.	Page 18, Line 6, is it the centre or the paediatrician that will be modelled as random effect? Or are they exactly the same?	We will be modelling the child's paediatrician as a random effect. The references to recruitment centre were included in error and have been removed (pg. 17).
36.	It seems the trial includes numerous outcomes, which will be compared at both 3M and 6M. Are you going to report p-values for all of these outcomes? Are you going to try to adjust for multiple testing? Is there some outcomes/time-points more important than others?	Yes, we will be reporting p values for all of these outcomes. We will not adjust for the primary outcome (so $p < .05$), but we will adjust for secondary outcomes to reduce Type I error inflation (e.g., through the False Discovery Rate method).
Economic Evaluation		
37.	Line 31: Will all the 15 outcomes (in table 2) really be compared for the costconsequences analysis? At which time point(s)?	The main type of analysis will be a cost-utility analysis. Rather than looking at all 15 outcomes, we have specified in the revised version that we will focus on the primary measure only, which will form a cost-effectiveness analysis (pg. 18). The inclusion of the CHU9D (Stevens, 2012) at baseline, 3-month and 6month follow-up will allow calculation of quality-adjusted life years (QALYs) as part of a cost-utility analysis, applying the area-under-the-curve method. A supplementary analysis which will be undertaken along with a costeffectiveness analysis will also be undertaken using the CSHQ as an outcome measure.
38.	Line 47: Clarify "any change". Does it mean the "difference" in resource use between baseline and the follow-up period will be analysed (how?).	We have removed 'any change' from the revised manuscript, as it was not our intention to compare resource use between baseline and the follow-up

		period. Instead, we will compare the costs between the two groups over the period of the study and adjust for cost differences at baseline. We have provided some further text in the revision (pg. 18).
39.	Line50: How will the costs and outcomes be compared between arms exactly (same model than for the outcome data analysis? Will it be adjusted?).	We have provided some further text in the revision (pp 19-20).
40.	Page19, Line17: Will Medicare and PBS collect data similar to the resource-use questionnaire? If so, which one will be used/how both information will be combined.	We have provided more information to clarify this in the revision (pg. 20).
Reviewer 5		
41.	Under aims and hypotheses, there are sections titled "Primary outcome" and "secondary outcomes" which are phrased like hypotheses. The authors should not conflate objectives/hypotheses with outcomes. These should be two separate sections.	The aims have been updated as requested (pg 8).
42.	I recommend clearly stating in the methods section whether the protocol conforms to SPIRIT reporting requirements.	This comment has been added to the overall study design subsection within the method (pg 10).
43.	The authors report that the study started in 2016. I suggest explicitly stating the stage the trial is at, at the time of submission. E.g. is recruitment ongoing or completed?	This has been updated as requested (pg 9). The recruitment of participants has finished since this protocol manuscript was submitted for publication.
44.	Under the "patient involvement" section: This section only refers to satisfaction with the intervention itself but not necessarily with the trial. Were patients/caregivers involved in designing the trial, e.g. selecting the most important research questions and/or outcomes or determining a minimally important difference (MID) for selected outcomes?	Children and caregivers were not involved in designing the trial. The trial was designed based on feedback of the intervention provided during the pilot study (Papadopoulos et al., 2015) and previous studies using the Sleeping Sound intervention in ADHD (Hiscock et al., 2015).
45.	Will children be asked for assent? if not, how will the investigators ensure that the children wish to participate (even if they cannot give full legal consent)?	The following has been provided to families: If you or your child finds any part of the research too challenging or anxiety provoking, you are able to take breaks, decline participation in any or all activities, or withdraw from the project. If any of the questions make you feel uncomfortable or you become distressed and you wish to speak to someone about this, please call a member of the research team on the details listed at the end of the Plain Language Statement. We also get explicit consent for Medicare/PBS consent for children that are 14 years or older.

46.	In the sample size section: Do the investigators have an estimate of the minimally important difference (MID) of their primary outcome? Is this sample size large enough to detect the MID?	Our sample size calculation is designed to detect a practically meaningful effect size (standardized mean difference of .5), which reflects: (1) the effect size estimated in our pilot work, whilst (2) recognising the possibility that this value may be lower in our full trial (this study).
47.	In the randomisation section: It looks like allocation concealment will not be an issue, but I recommend commenting explicitly on allocation concealment.	We have added a specific comment as suggested (pg 12).
48.	The randomisation section states that siblings will be allocated to the same intervention. What percentage of families are expected to enroll multiple children? This becomes a sort of cluster trial where the unit of randomization is a family. I recommend explicitly stating the unit of randomization in this paper. It may have an impact on the statistical methodology required depending on how many children are enrolled this way. If a single family enrolls two children, does this count as two participants toward enrollment or one? This will affect sample size and needs to be clarified.	This is very unlikely to happen. We have added a section to the analysis subsection that directly addresses analytic solutions to evaluate any cluster effects that may result. We do not formally include a family cluster into the model because the anticipated numbers of >1 participant per family is likely to be low, causing convergence issues in the model and unreliable parameter estimates for our mixed models. We will instead conduct some sensitivity analyses to evaluate the size and likely impact of any clustering should it arise (all covered in the revised manuscript – pp 17-18).
49.	How will co-interventions be minimized/dealt with? It would be unlikely that TAU participants would fully cross-over into the treatment group but they may be trying similar interventions on their own, or supplements, medications etc that may have some effect on sleep.	Data pertaining to the use of both medications and psychology services has been collected at baseline and also at the 3- and 6- month follow ups. We have also included the lack of data collected around engaging in sleep treatments post enrolment as a limitation of the study (pg 3).
50.	The SPIRIT statement requires an explanation of data collection/management processes, quality assurance/monitoring, a description of how participant retention will be maximized/addressed, any safety monitoring, interim analyses planned, and whether/how harms will be addressed. I recommend that the authors add these descriptions and double-check that all SPIRIT statement items are explicitly addressed in the protocol (spirit-statement.org).	More detail about the secure online database for data collection has been included, and assessment of fidelity to the intervention. We have also addressed all of the SPIRIT guidelines.
51.	Only one limitation is mentioned in the point form strengths and limitations section. The authors should discuss whether there are likely to be any more limitations and how they will be/have been addressed either by design or analysis.	Additional limitations have been added to the strengths and limitations section of the manuscript (pg 5. We have suggested a number of supplementary analysis to account for covariates that may impact our results. Please see statistical analysis section (pp 17-18).

VERSION 2 – REVIEW

REVIEWER	Sela Sanberg, Ph.D., BCBA-D Western Michigan University, U.S.A.
REVIEW RETURNED	14-Jun-2019

GENERAL COMMENTS	I would like to acknowledge the authors efforts toward improving the originally submitted manuscript. In the revised manuscript, the authors included additional citations to support their literature review, as well as addressed noted limitations and provided details needed for study replication. However, improvements are still needed with regard to meeting the APA Publication Manual standards. Therefore, my recommendation is for minor revisions to further prepare the manuscript for publication. Below are examples of items for improvement, which should be addressed while referencing the APA Publication Manual for details. Please note that this is not a complete list of all necessary changes to improve the quality of the manuscript, however, these suggestions for improvement may be helpful and applied throughout the manuscript. On page 1, line 6, include (RCT) after the term 'randomized control trial' and replace 'autism spectrum disorder' with ASD abbreviation. The abbreviation ASD was already introduced in the sentence above. Page 2, line 54, include the abbreviation (CSHQ) after the introduction of the term Children's Sleep Habits Questionnaire. Page 3, line 36, if the measure has yet to be introduced, include the full title followed by the abbreviation for both the ADOS-2 and ADI-R. Page 4, lines 22-23, capitalize Attention-Deficit Hyperactivity Disorder. Eliminate the word 'very' and the use of any other intensifiers in academic or scientific writing. Check on the use of a small tilde in APA formatting and style guidelines. Page 4, second paragraph, more recent references are available and needed. Additionally, identify all sleep problems experienced by children with ASD. For example, studies looking at undesired co-sleeping and dependent sleep initiation in ASD should be included and correctly cited. See Sanberg et al., 2018 for information on sleep variables and a citation. For the last paragraph on page 4, please provide a more comprehensive account of biopsychosocial factors that impact sleep in children with ASD. See and cite Johnson et al., 2018; and Levin et al., 2016. Biological factors include more than "melatonin secretion". Perhaps look into serotonin, GABA, nocturnal cortisol. Explain what you mean by "sleep cues". Do you mean stimuli? In addition, "sleep cues" do not establish "healthy sleep-wake cycles." Engaging in specific behavior (e.g., learning healthy sleep habits) is how healthy sleep-wake cycles are established and maintained. On page 5, line 6, please specify what you mean by "family factors". Are you referring to maternal stress, paternal stress, or
---

	marital stress? Sibling conflict? Concomitant autonomic arousal? See Hodge et al., 2014. Please adhere to the APA guideline with regard to spacing, grammar, punctuation, citations, and references. For example, on page 28, items 4, 5, 6, 7, and 8 are not correctly referenced according to APA. Great examples are provided in the APA Publication Manual. Best wishes as the study concludes. I look forward to reading this publication and any future publications that include the results.
--	--

REVIEWER	Baptiste Leurent LSHTM, UK
REVIEW RETURNED	19-Jun-2019

GENERAL COMMENTS	Thank you for your revisions, the manuscript has improved in several aspects. My main concern remaining is about the sample size paragraph, which I don't think is clear enough to be reproduced. Main comment: Thank you for providing more information regarding the sample size calculation. However, I feel like the current description still does not allow to reproduce fully the sample size calculation. For example, was the sample size calculated for a single time-point (and if so why was a within-person clustering included? Or was it to take into account of adjustment for baseline, if so, it should be indicated). If the multiple follow-ups were taken into account (at 3 and 6M), what treatment effect was assumed at 6M? Adjusting for baseline? Or I am also still not clear what distribution was assumed for the cluster size. In my view, the sample size was maybe unnecessarily complex for the trial design, but this would be fine as long as it is clearly described. I would advise to provide in appendix the code used for the simulations, which would allow to understand more clearly the analysis model, and parameters assumed. Or the Gpwer output, with all the parameters clearly listed. Minor comments: -- Sample size -- Page 16, Line 40 (revised version without track cahnges): please provide the CHSQ equivalent in mean difference, not %. Also indicate the expected standard deviation for CSHQ. Page 16, Line 57: You indicated the ICC sources in your responses, but these should be included in the manuscript as well. P17, L5: "<10% bias for model coefficients and <5% bias for standard errors" is not clear. What does this "<10% bias" means? Your analysis should be unbiased, but not sure how you quantified "sufficient precision". Consider removing? I suspect the achieved power is the main information here. -- Data analysis -- The data analysis is now much clearer, but maybe still not specific enough regarding the primary analysis.
--

	Page 17: specify somewhere that the primary outcome time-point will be at 3-month. This should also be in the abstract P17, L47. Need to be more specific about how treatment effect will be modelled in the analysis. I presume will not allow for a group (treatment arm) effect at baseline, then allow for 2 different group effects at 3M and 6M? P17, L58: Will the adjusted or unadjusted analysis be primary? P18,L22 : I guess should read “missingness can be predicted from the variables...” not “cannot”. P18: you should indicate somewhere how you will address the multiple-testing. You answered in the reviewers’ responses but did not include it in the manuscript. -- Economic evaluation -- This section is now much clearer, and I do not have any further comments.
--	---

REVIEWER	Kim Madden McMaster University, Canada
REVIEW RETURNED	11-Jun-2019

GENERAL COMMENTS	The authors have addressed all of my comments sufficiently.
---

VERSION 2 – AUTHOR RESPONSE

Reviewer 1							
Reviewer Name: Sela Sanberg, Ph.D., BCBA-D Institution and Country: Western Michigan University, U.S.A.							
	   Reviewer comments Authors response     1. Below are examples of items for improvement, which should be addressed while referencing the APA Publication Manual for details. Please note that this is not a complete list of all necessary changes to improve the quality of the manuscript, however, these suggestions for improvement may be helpful and applied throughout the manuscript Thank you for these suggestions. We have revised the manuscript for clarity and consistency of formatting.   2. On page 1, line 6, include (RCT) after the term 'randomized control trial' and replace 'autism spectrum disorder' with ASD abbreviation. The abbreviation APA guidelines recommend avoiding the use of abbreviations in the title. We have included the abbreviation in the first sentence as it refers to the proper name of   	Reviewer comments	Authors response	1. Below are examples of items for improvement, which should be addressed while referencing the APA Publication Manual for details. Please note that this is not a complete list of all necessary changes to improve the quality of the manuscript, however, these suggestions for improvement may be helpful and applied throughout the manuscript	Thank you for these suggestions. We have revised the manuscript for clarity and consistency of formatting.	2. On page 1, line 6, include (RCT) after the term 'randomized control trial' and replace 'autism spectrum disorder' with ASD abbreviation. The abbreviation	APA guidelines recommend avoiding the use of abbreviations in the title. We have included the abbreviation in the first sentence as it refers to the proper name of
Reviewer comments	Authors response						
1. Below are examples of items for improvement, which should be addressed while referencing the APA Publication Manual for details. Please note that this is not a complete list of all necessary changes to improve the quality of the manuscript, however, these suggestions for improvement may be helpful and applied throughout the manuscript	Thank you for these suggestions. We have revised the manuscript for clarity and consistency of formatting.						
2. On page 1, line 6, include (RCT) after the term 'randomized control trial' and replace 'autism spectrum disorder' with ASD abbreviation. The abbreviation	APA guidelines recommend avoiding the use of abbreviations in the title. We have included the abbreviation in the first sentence as it refers to the proper name of						

	ASD was already introduced in the sentence above.	the sleep intervention trial, rather than denoting the use of ASD as a consistent abbreviation for use in the article.
3.	Page 2, line 54, include the abbreviation (CSHQ)after the introduction of the term Children's Sleep Habits Questionnaire.	Thank you, this has been added.
4.	Page 3, line 36, if the measure has yet to be introduced, include the full title followed by the abbreviation for both the ADOS-2 and ADI-R.	Thank you, this has been added.
5.	Page 4, lines 22-23, capitalize Attention-Deficit Hyperactivity Disorder.	We have written attention-deficit/hyperactivity disorder (ADHD)according to APA style format without capitalisation and consistent with other BMJ Open articles.
6.	Eliminate the word 'very' and the use of any other intensifiers in academic or scientific writing.	Thank you, we have updated the manuscript accordingly.
7.	Check on the use of a small tilde in APA formatting and style guidelines.	We have removed the use of the small tilde.
8.	Page 4, second paragraph, more recent references are available and needed.	Thank you. Additional updated references have been added
9.	Additionally, identify all sleep problems experienced by children with ASD. For example, studies looking at undesired co-sleeping and dependent sleep initiation in ASD should be included and correctly cited. See Sanberg et al., 2018 for information on sleep variables and a citation.	This section has been extended and additional references have been added.
10.	For the last paragraph on page 4, please provide a more comprehensive account of biopsychosocial factors that impact sleep in children with ASD.	This section has been extended and additional references added.
	See and cite Johnson et al., 2018; and Levin et al., 2016. Biological factors include more than "melatonin secretion". Perhaps look	

	into serotonin, GABA nocturnal cortisol.	
11.	Explain what you mean by "sleep cues". Do you mean stimuli? In addition, "sleep cues" do not establish "healthysleep-wake cycles." Engaging in specific behavior (e.g., learning healthy sleep habits) is how healthy sleepwake cycles are established and maintained.	Thank you for bringing this awkward phrasing to our attention. We have clarified the phrasing to reduce confusion for the reader.
12.	On page 5, line 6, please specify what you mean by "family factors". Are you referring to maternal stress, paternal stress, or marital stress? Sibling conflict? Concomitant autonomic arousal? See Hodge et al., 2014.	This section has been extended and additional references added.
13.	Please adhere to the APA guideline with regard to spacing, grammar, punctuation, citations, and references. For example, on page 28, items 4, 5, 6, 7, and 8 are not correctly referenced according to APA. Great examples are provided in the APA Publication Manual. Best wishes as the study concludes. I look forward to reading this publication and any future publications that include the results.	We have adhered to BMH journal guidelines. BMJ journals use a modified Vancouver approach for citations and references rather than an APA style.
Reviewer 4		
Reviewer Name: Baptiste Leurent Institution and Country: LSHTM, UK		
Sample Size		

14.	My main concern remaining is about the sample size paragraph, which I don't think is clear enough to be reproduced. Thank you for providing more information regarding the sample size calculation. However, I feel like the current description still does not allow to reproduce fully the sample size calculation. For example, was the sample size calculated for a single time-point (and if so why was a within-person clustering included? Or was it to take into account of adjustment for baseline, if so, it should be indicated). If the multiple follow-ups were taken into account (at 3 and 6M), what treatment effect was assumed at 6M? Adjusting for baseline? Or I am also still not clear what distribution was assumed for the cluster size. In my view, the sample size was maybe unnecessarily complex for the trial design, but this would be fine as long as it is clearly described.	We have heavily revised the paragraph that details sample size calculations, emphasizing that the study sample size was powered on a standardized mean difference, and that the primary outcome of interest is at 3-month follow-up. We also emphasize that the primary analysis is the unadjusted model, and we test the robustness of results subsequently with an adjustment for covariates. We have now provided a supplementary excel file with the calculations that lead to our final sample size of 234. With respect to the clustering effect, we include this because we are using a linear mixed effects model with baseline scores covaried, and hence within-person clustering for the repeated measures aspect of this model. In the sample size section, we now add explicit mention that the within person ICC is to account for this repeated measurement. Given that we provide a conservative estimate of effect size for the 3M follow-up and assume that treatment benefits
-----	---	--

		are maintained at 6M, we assume a similar effect size at the 6M follow-up. We hope that this added information now makes the sample size and analytic decisions clearer and replicable.
15.	I would advise to provide in appendix the code used for the simulations, which would allow to understand more clearly the analysis model, and parameters assumed. Or the Gpower output, with all the parameters clearly listed.	Sorry that this was still unclear. We have heavily revised this passage, providing the formula for required sample size for a between-group mean difference, and explanation of the subsequent adjustments for attrition and clustering. We then

					provide these calculations for external verification as an excel file.
16.	Page 16, Line 40 (revised version without track changes): please provide the CHSQ equivalent in mean difference, not %. Also indicate the expected deviation for CSHQ.				We have moved reference to % change. We have emphasized that the effect size is a standardized mean difference of .5, with a standard deviation of 1. As the reviewer pointed out in several comments, there were pieces of information provided in the earlier version of the manuscript that were extraneous and made it confusing for the reader to understand what was used for sample size calculations. We have removed unnecessary content and refined to essentials to enable the reader to replicate our sample size calculation.
17.	Page 16, Line 57: You indicated the ICC sources in your responses, but these should be included in the manuscript as well.				Thank you, we have reference this in the manuscript.
Data Analysis					
18.	P17, L5: “<10% bias for model coefficients and <5% bias for standard errors” is not clear. What does this “<10% bias” mean? Your analysis should be unbiased, but not sure how you quantified “sufficient precision”. Consider removing? I suspect the achieved power is the main information here.				Thank you for this suggestion – these terms and cutoffs for bias are commonly mentioned in this way when implementing simulation studies in Mplus. However, we can see that it adds confusion and isn’t particularly helpful. Hence, we have removed as per suggestion.
19.	The data analysis is now much clearer, but maybe still not specific enough regarding the primary analysis. Page 17: specify somewhere that the primary outcome time-point will be at 3-month. This should also be in the abstract				We have added mention in the abstract, sample size and data analysis subsections that the primary outcome timepoint is 3-months post randomization.

20.	P17, L47. Need to be more specific about how treatment effect will be modelled in the analysis. I presume will not allow for a group (treatment arm) effect at baseline, then allow for 2 different group effects at 3M and 6M?	That is correct. We have added an explicit statement in the revised Data Analysis section stating we have separate parameters to estimate group differences at 3 and 6M.
21.	P17, L58: Will the adjusted or unadjusted analysis be primary?	The unadjusted results will be primary, and we will include adjusted results to evaluate the robustness of these results.
22.	P18,L22 : I guess should read "missingness can be predicted from the variables..." not "cannot".	Thank you for bringing this awkward phrasing to our attention. We have removed the phrasing to reduce confusion for the reader.
23.	P18: you should indicate somewhere how you will address the multiple testing. You answered in the reviewers' responses but did not include it in the manuscript.	We now include a statement reflecting this strategy at the end of the second last paragraph of the Data Analysis subsection of the manuscript.

VERSION 3 – REVIEW

REVIEWER	Sela Sanberg, Phd., BCBA-D Western Michigan University Oregon Health & Science University
REVIEW RETURNED	12-Aug-2019

GENERAL COMMENTS	I would like to applaud the authors on improving their initial manuscript by incorporating reviewers' feedback and addressing prior concerns regarding replicability. In the current revision, the authors have updated their literature review, provided a more compelling argument to support their study, utilized established sleep terminology, and clarified the methodology of the study. Therefore, my recommendation is to accept this manuscript for publication in BMJ Open. I look forward to learning about the effectiveness of this brief behavioral intervention in children with Autism Spectrum Disorder (ASD) and sleep problems within the Australian Health Care System.
---

REVIEWER	Baptiste Leurent LSHTM, UK
REVIEW RETURNED	20-Aug-2019

GENERAL COMMENTS	Thank you for this revised version. All my concerns regarding the sample size and analysis have now been addressed, and recommend this paper for publication.
--

	Note that I believe some errors were made in the sample size calculations. For example, you used a design effect formula for a cluster-randomised trial, which is not the same as for an individually-randomised trial with clustering. As an example, with the corrections you made, the more repeated-measurements done, the larger the sample size should be, which is counter-intuitive. However, I believe the final sample size is reasonable (if anything may be too conservative by applying these clustering corrections). Now that the formula and code have been provided, it is sufficiently transparent and reproducible, which was my main concern.
--	--